# Self-Management Interventions for Adults Living with Type II Diabetes to Improve Patient-Important Outcomes: An Evidence Map

**DOI:** 10.3390/healthcare11243156

**Published:** 2023-12-13

**Authors:** Yang Song, Jessica Beltran Puerta, Melixa Medina-Aedo, Carlos Canelo-Aybar, Claudia Valli, Marta Ballester, Claudio Rocha, Montserrat León Garcia, Karla Salas-Gama, Chrysoula Kaloteraki, Marilina Santero, Ena Niño de Guzmán, Cristina Spoiala, Pema Gurung, Fabienne Willemen, Iza Cools, Julia Bleeker, Rune Poortvliet, Tajda Laure, Marieke van der Gaag, Kevin Pacheco-Barrios, Jessica Zafra-Tanaka, Dimitris Mavridis, Areti Angeliki Veroniki, Stella Zevgiti, Georgios Seitidis, Pablo Alonso-Coello, Oliver Groene, Ana Isabel González-González, Rosa Sunol, Carola Orrego, Monique Heijmans

**Affiliations:** 1Institut de Recerca Sant Pau (IR SANT PAU), Sant Quintí 77-79, 08041 Barcelona, Spain; yangsongcochrane@gmail.com (Y.S.); beltranpuerta.jessica@gmail.com (J.B.P.); mmedinaa@santpau.cat (M.M.-A.); carlos.canelo.ay@gmail.com (C.C.-A.); claudia.valli89@gmail.com (C.V.); claudiorochacalderon@gmail.com (C.R.); mleong@santpau.cat (M.L.G.); ksalasgama@gmail.com (K.S.-G.); marilinasantero@gmail.com (M.S.); ena.ninodeguzman@gmail.com (E.N.d.G.); palonso@santpau.cat (P.A.-C.); 2Avedis Donabedian Research Institute (FAD), 08037 Barcelona, Spain; mballester@fadq.org (M.B.); kevin.pacheco.barrios@gmail.com (K.P.-B.); j.zafra.t@gmail.com (J.Z.-T.); aigonzalez@fadq.org (A.I.G.-G.); rsunol@fadq.org (R.S.); corrego@fadq.org (C.O.); 3Faculty of Medicine, Universitat Autὸnoma de Barcelona (UAB), 08025 Barcelona, Spain; 4Network for Research on Chronicity, Primary Care, and Health Promotion (RICAPPS), 28029 Madrid, Spain; 5Netherlands Institute for Health Services Research (NIVEL), 3513 Utrecht, The Netherlands; cristina_spoiala@yahoo.co.uk (C.S.); pemagurung2017@yahoo.com (P.G.); fabiennewillemen@live.nl (F.W.); izacools@hotmail.com (I.C.); j.bleeker@live.nl (J.B.); runepoortvliet@gmail.com (R.P.); tajda.laure91@gmail.com (T.L.);; 6Department of Primary Education, School of Education, University of Ioannina, 45110 Ioannina, Greece; dimi.mavridis@googlemail.com (D.M.); stella.zevgiti@gmail.com (S.Z.); g.seitidis@uoi.gr (G.S.); 7Knowledge Translation Program, Li Ka Shing Knowledge Institute, St. Michael’s Hospital, Unity Health Toronto, Toronto, ON M5B 1W8, Canada; areti-angeliki.veroniki@unityhealth.to; 8Institute for Health Policy, Management, and Evaluation, University of Toronto, Toronto, ON M5S 3G8, Canada; 9Centro de Investigación Biomédica en Red de Epidemiología y Salud Pública (CIBERESP), 28029 Madrid, Spain; 10OptiMedis, 20095 Hamburg, Germany; o.groene@optimedis.de; 11Faculty of Management, Economics and Society, University of Witten/Herdecke, 58455 Witten, Germany

**Keywords:** diabetes type 2, self-management interventions, evidence mapping, quality improvement

## Abstract

Self-management interventions (SMIs) may be promising in the treatment of Diabetes Mellitus Type 2 (T2DM). However, accurate comparisons of their relative effectiveness are challenging, partly due to a lack of clarity and detail regarding the intervention content being evaluated. This study summarizes intervention components and characteristics in randomized controlled trials (RCTs) related to T2DM using a taxonomy for SMIs as a framework and identifies components that are insufficiently incorporated into the design of the intervention or insufficiently reported. Following evidence mapping methodology, we searched MEDLINE, CINAHL, Embase, Cochrane, and PsycINFO from 2010 to 2018 for randomized controlled trials (RCTs) on SMIs for T2DM. We used the terms ‘self-management’, ‘adult’ and ‘T2DM’ for content. For data extraction, we used an online platform based on the taxonomy for SMIs. Two independent reviewers assessed eligible references; one reviewer extracted data, and a second checked accuracy. We identified 665 RCTs for SMIs (34% US, 21% Europe) including 164,437 (median 123, range 10–14,559) adults with T2DM. SMIs highly differed in design and content, and characteristics such as mode of delivery, intensity, location and providers involved were poorly described. The majority of interventions aimed to improve clinical outcomes like HbA1c (83%), weight (53%), lipid profile (45%) or blood pressure (42%); 27% (also) targeted quality of life. Improved knowledge, health literacy, patient activation or satisfaction with care were hardly used as outcomes (<16%). SMIs most often used education (98%), self-monitoring (56%), goal-setting (48%) and skills training (42%) to improve outcomes. Management of emotions (17%) and shared decision-making (5%) were almost never mentioned. Although diabetes is highly prevalent in some minority groups, in only 13% of the SMIs, these groups were included. Our findings highlight the large heterogeneity that exists in the design of SMIs for T2DM and the way studies are reported, making accurate comparisons of their relative effectiveness challenging. In addition, SMIs pay limited attention to outcomes other than clinical, despite the importance attached to these outcomes by patients. More standardized and streamlined research is needed to better understand the effectiveness and cost-effectiveness of SMIs of T2DM and benefit patient care.

## 1. Introduction

With the aging of populations worldwide, chronic conditions are a major concern, given their significant impact on individual patients, health care and society as a whole. The World Health Organization (WHO) estimates that by 2025, chronic diseases will account for 73% of all deaths and 60% of the global disease burden [1]. 

One chronic disease that has rapidly evolved during the last decades as a major public health problem is Type 2 Diabetes Mellitus (T2DM). The global prevalence of T2DM in adults was about 536.6 million people in 2021, and this number is expected to increase further to 783.2 million people by 2045 [2]. Diabetes is a chronic metabolic disease characterized by dysregulation of carbohydrate, lipid and protein metabolism, and results from impaired insulin secretion, insulin resistance or a combination of both. The management of T2DM often involves a combination of medical treatments and lifestyle changes geared to normalize blood sugar levels and decrease cardiovascular risk. These may include medication, diet, exercise, and regular self-monitoring activities, of which the success ultimately relies on patients’ abilities to accept and take responsibility for their disease [3]. For most patients, this self-management is a difficult task that challenges them on a daily basis and often has a considerable impact on work, family and social life [4,5]. 

Self-management interventions (SMIs) are developed to support people in their daily self-management tasks. Although different definitions of SMIs exist [6], in general, SMIs can be characterized as supportive interventions that healthcare staff, peers, or laypersons provide to increase patients’ skills and confidence in managing their long-term condition. Interventions to support self-management of T2DM may include, among others, education, support for self-monitoring, lifestyle advice, goal setting for behavioral change and coaching [7]. 

Evidence has shown that SMIs for type 2 diabetes can be effective, for example, by reducing glycated hemoglobin (HbA1c) and by losing weight [8,9]. However, it remains unclear which components or approaches to self-management support contribute most to this effectiveness [6,8]. This is mainly due to the heterogeneity in study design and reporting [10]. Heterogeneity also hinders the knowledge translation from scientific evidence into clinical practice and the replication of successful SMIs by other researchers. The aim of the present study is to summarize intervention components in RCTs related to self-management for T2DM and identify components that are insufficiently incorporated or insufficiently reported into the design of the intervention. By detecting shortcomings either in the design or in the description of SMIs for T2DM, we aim to provide a foundation for the design of new interventions in diabetes self-management, easier replication of effective SMIs, and contribute to the improvement of the future reporting of interventions. In this way, we facilitate the replication and build on research findings for other researchers and strengthen the evidence base.

## 2. Materials and Methods

This study was conducted as part of a larger study, COMPAR-EU (see Box 1), and uses the descriptive data on T2DM collected there.

Box 1The COMPAR-EU Project.COMPAR-EU was a multimethod, interdisciplinary project that contributed to bridging the gap between current knowledge and practice of self-management interventions. COMPAR-EU aimed to identify, compare, and rank the most effective and cost-effective self-management interventions (SMIs) for adults in Europe living with one of the four high-priority chronic conditions: type 2 diabetes, obesity, chronic obstructive pulmonary disease and heart failure. The project provides support for policymakers, guideline developers and professionals to make informed decisions on the adoption of the most suitable self-management interventions through an IT platform featuring decision-making tools adapted to the needs of a wide range of end users (including researchers, patients, and industry).COMPAR-EU launched in January 2018 and was completed in December 2022, contributing the following outputs: (i) an externally validated taxonomy composed of 132 components, classified in four domains (intervention characteristics, expected patient (or carer) self-management behaviors, type of outcomes and target population characteristics), (ii) Core Outcome Sets (COS) for each disease, including 16 outcomes for COPD, 16 for Heart Failure, 13 for T2DM and 15 for Obesity, (iii) extraction and descriptive results for each disease based on 665 studies for Diabetes, 252 studies for COPD, 288 studies for Heart Failure and 517 studies for Obesity, (iv) comparative effectiveness analysis based on a series of pairwise meta-analyses, Network Meta Analysis (NMAs) and component NMAs (CNMA) for all outcomes across all four diseases, (v) contextual analysis addressing information on equity, acceptability and feasibility; general information on contextual factors on the level of patients, professionals, their interaction and the health care organization for those interested in implementation, (vi) cost effectiveness conceptual models have been created for each chronic condition including risk factors or intermediate variables relevant for SMIs and final outcomes, (vii) business plans and a sustainability strategy was developed based on a multi-prong approach including qualitative interviews with managers and clinicians, the focus group with clinical representatives from EU countries, workshops with industry representatives and a hackathon event. The majority of the COMPAR-EU end-products are available on the online COMPAR-EU platform: www.self-management.eu.

For the aim of the present study, summarizing descriptive data of RCTs on SMIs in T2DM, we followed the methodology of evidence mapping [11]. An evidence map, like a scoping review, is an evidence synthesis methodology that addresses broad research questions, aiming to describe a bigger picture than systematic reviews. An evidence map provides a comprehensive overview of the available evidence on a specific topic by describing and highlighting its characteristics and gaps [12]. Evidence mapping not only provides an overview of the available literature but also graphically illustrates what evidence is available [13]. Having a clear overview of the available evidence is important to understand the coverage of the literature and to develop suggestions and strategies for future research. The user-friendly format of evidence maps facilitates communication with various stakeholders, such as policymakers, researchers, and health professionals [14].

We reported this evidence map following the PRISMA Extension for Scoping Reviews (PRISMA-ScR), which is recommended for evidence mapping [15]. The review methodology was established before data extraction and was registered in PROSPERO (CRD42020155441) and published [16].

### 2.1. Literature Search 

To identify relevant RCTs for T2DM, we followed two steps: first, we searched the databases of a previous European project, PRO-STEP [17], that had identified systematic reviews on SMIs for T2DM published from 2000 up to 2015. Then, we updated this data set with a search strategy in PubMed, CINAHL, Embase, Cochrane and PsycINFO from 2010 to 2018 (see Appendix A). The updated search overlaps in time with the PRO-STEP data set to reduce the chance of missing any eligible studies. The updated search was guided by the search string used for PRO-STEP and consisted of key terms regarding ‘self-management’, ‘adults’, ‘diabetes’ for content and ‘randomised controlled trial’ for study type. MESH terms were used to find relevant synonyms for all key terms. To take diversity into account, this search string was repeated with the addition of terms related to comorbidity, gender, minority groups, socio-economic status (SES) and health literacy. The final search was peer-reviewed by a librarian expert (IS) and adapted to the requirements of each database [18].

### 2.2. Screening and Selection Process

For the screening process of titles and abstracts, the following inclusion criteria were used: (i) population, adults (≥18 years of age) with a diagnosis of T2DM and/or their caregivers. When a study had a mixed population of patients (i.e., not only people with T2DM) and did not report the outcomes per condition separately, the study was included if at least 80% of the population targeted T2DM; (ii) Intervention, SMIs; (iii) Comparison, usual care (usual care as indicated by the authors), other SMIs (head-to-head); (iv) Outcomes, studies had to report at least one of the outcomes from the “COMPAR-EU” predefined COS; this COS was developed together with T2DM patients and healthcare professionals and contained those outcomes that were rated as most important to them in the context of self-management. (v) Study design: RCTs; quasi-randomized studies were excluded. We included only studies published in English or Spanish.

The titles and abstracts of the retrieved studies were screened by pairs of two independent researchers. Before the screening, all researchers had to perform a calibration exercise to reach a gold standard proposed by a team of experienced supervisors (M.H., M.B., C.C.-A., or C.O.). Each reviewer judged 40 studies for inclusion based on title and abstract. Doubts were discussed with an experienced third researcher (M.H.). All reviewers had to reach at least 80% agreement with the gold standard (calibration) before starting the remaining of the title/abstract screening. Titles and abstracts of the retrieved studies were screened using Covidence (www.covidence.org). For the full-text screening, the same calibration exercise was performed for 20 full-text papers. The final eligibility of each included full-text study judged by two independent researchers was confirmed by a supervisor (M.H., M.B., C.C.-A., or C.O.) through consensus. 

### 2.3. Data Extraction and Collection 

Data were extracted using a standard data extraction form that was embedded in an online platform and that was developed in earlier stages of the project (https://platform.self-management.eu/randomized-controlled-trials (accessed on 23 June 2023)). One reviewer extracted relevant data from eligible studies, and a second reviewer checked accuracy. The data extraction form followed the COMPAR-EU taxonomy structure of SMIs (Figure 1). The data extraction form included patient characteristics, including comorbidities, gender and socio-economic variables (e.g., health literacy), SMI support techniques and expected self-management behaviors [10], outcomes, and information on study design and risk of bias. The support techniques contained 12 methods to support self-management: sharing information, skills training, self-monitoring, prompt use, goal setting, problem-solving, coaching, emotional management, social support, shared decision making, service use and provision of equipment. For each intervention, delivery methods, type of encounters, total time devoted to self-management support, location, and type of provider were also extracted. The targeted outcomes consisted of 23 outcomes in the core outcome set for T2DM developed during COMPAR-EU and included outcomes related to basic empowerment skills (self-efficacy, knowledge, patient activation and health literacy); outcomes related to adherence to self-management behaviors in general and more specific to physical activity, diet, medication, smoking cessation and self-monitoring activities; clinical outcomes such as HbA1c, weight, lipid profile, blood pressure, hypoglycemia, hyperglycemia, complications and life expectancy; quality of life, participation in decision making, satisfaction with care and use of scheduled and unscheduled care. If the same intervention and/or the results were reported in multiple papers, we combined them in the same extraction form as one study to obtain the most complete extraction possible. In addition to the descriptives, the risk of bias was assessed using Cochrane’s tool for assessing the risk of bias in included studies, and then a second reviewer verified the judgments [19]. When there was not enough information reported to decide, we contacted the study authors and requested clarification or further information. We rated the risk of bias as low risk, high risk, or unclear for all included studies in each of the five domains of the Cochrane ‘Risk of bias’ tool.

### 2.4. Data Analysis

We summarized the study findings through descriptive analysis. 

## 3. Results

### 3.1. Search Results

A total of 11,780 citations were identified by the searches, including primary studies from PRO-STEP, PubMed, Embase, CINAHL, PsycINFO and Cochrane (see Figure 2 and Appendix A). Following the title and abstract screening, 1657 full-text reports were reviewed. Eventually, 1026 articles were uploaded to the COMPAR-EU platform for data extraction. From these, 665 studies were finally included for the descriptive analysis. The main reasons for exclusion during the extraction phase (n = 361) were: being multiple publications from included studies (n = 125), no outcomes from COS (n = 72), no SMI (n = 68), no RCT (n = 45), or wrong population (n = 51). 

### 3.2. Key Characteristics of Included RCTs

The 665 studies, composing 164,437 T2DM adults with a median number of 123 adults per RCT (range 10–14,559) and a median age of 58 years old, were conducted in 64 different countries; 141 were conducted in Europe (21%), 79% outside Europe and only five studies were conducted in more than one country. Most of the studies came from the United States (35%), followed at a distance by Iraq (7%), the United Kingdom (6%), China (6%) and Korea (5%) (Table 1). Almost all studies were implemented on an individual patient level (92%) as compared to the population level. Almost without exceptions, studies were developed for patients (99%), with only one study for caregivers and eight targeting both patients and caregivers (Table 1).

The number of intervention arms (n = 879 in total) in these 665 studies varied between two and five, but the majority of the studies (90%) included two arms. Most studies compared an SMI to usual care (n = 530, 80%), whereas 135 studies compared one or more intervention arms (head-to-head interventions). Usual care was defined as such by the authors and included regular visits and a form of education in most cases. In some studies, usual care (as indicated by the authors) consisted of something more than just information or education, for example, skills training or coaching. In this case, we called it ‘usual care plus’. In 20% of all intervention arms (n = 879), the intervention content or delivery methods were tailored to the characteristics of the study population (e.g., educational material of an existing intervention that was simplified because of respondents with low health literacy, translated because of Spanish speaking people, or adapted because of known gender differences between men and women).

### 3.3. Characteristics of the Participants

Across all studies, participants were more often female (mean 57%, SD 49–67%); the mean age was 58 years old, and the mean time since diagnosis was 8.6 years across studies. Although most studies used general samples of T2DM, others used more specific inclusion criteria: 72 studies (11%) focused specifically on populations with a low socio-economic status. In most of these studies, education or income was used as a proxy for inclusion; 13% of the studies targeted specific minority groups. These were mostly studies from the United States and concerned immigrants in general or more specific groups such as African Americans, Mexican Americans, Latinos or veterans. Information on health literacy levels was only provided in 29 studies (4%); 13% (n = 88) of the studies focused on diabetes patients with comorbidity; among them, 22% did not specify the type of comorbidity. In studies that did, obesity, hypertension and depression were the most common comorbidities.

Most studies described their T2DM populations with respect to sex (96%), age (97%) and diabetes control (HbA1c; 83%). Other information on study populations, such as illness duration, comorbidities, socio-economic status characteristics and health literacy levels, were described less frequently. Age (61% of trials used an age range for patients to be included) and diabetes control (41% of the studies used a threshold value for HbA1c) were most often used as specific inclusion criteria. Other characteristics, such as time since diagnosis (17%), belonging to a cultural minority group (11%), having comorbidity (9%), sex (4%), socio-economic status (4%) and health literacy levels (1%), were less often used as explicit inclusion criteria for the SMIs in diabetes.

### 3.4. Characteristics of the SMIs Reported 

Table 2 shows the frequency in which specific SM support techniques were used in the intervention arms as well as the frequency of expected self-management behaviors the intervention arms focused on. Figure 3 shows a matrix with the frequency with which specific SM support techniques are combined across studies. Figure 4 shows the number of studies in which expected behaviors go together in one study.

#### 3.4.1. Self-Management Support Techniques 

Self-management techniques are techniques or methods used to provide care and encouragement to people with chronic conditions and their carers to help them understand their central role in managing their condition, make informed decisions about care and engage in appropriate behaviors. In the intervention arms, the number of self-management support techniques varied between 1 and 11 (median 4, IQR 3–6). Sharing information was used in almost all intervention arms (98%), followed by self-monitoring (56%), goal setting (48%) and skills training (42%). Other techniques, for example, learning skills to handle emotions and learning to use social support or external resources, such as specific websites, were reported in less than a fifth of the studies. Shared decision-making was least mentioned as a technique to support self-management (5%) (Table 2). It appeared that in the 879 intervention arms, a specific combination of support techniques was frequently offered: sharing information plus self-monitoring (n = 488), sharing information plus goal setting (n = 418), sharing information plus skills training (n = 359), sharing information plus problem-solving (n = 308) and self-monitoring plus goal setting (n = 279) (Figure 3).

#### 3.4.2. Expected Self-Management Behaviors

Expected self-management behaviors refer to decisions and behaviors that patients with chronic diseases are expected to engage in to improve their health. These behaviors are the focus of the self-management interventions and support techniques. In the intervention arms, the number of expected behaviors mentioned varied between 1 and 12 (median 3, IQR 2–5). Expected behaviors of T2DM patients most often included healthy eating (62%) and physical activity (61%), both being lifestyle-related behaviors; self-monitoring (63%), condition-specific behaviors like checking your feet (48%) and medication use (41 were also frequently mentioned; behaviors in relation to work and social roles, healthy sleep, alcohol or smoking reduction and communicating with health care were seldomly reported (Figure 4). Figure 4 also shows that in the 879 intervention arms, the combinations of healthy eating and physical activity (n = 474), healthy eating and self-monitoring (n = 390) and physical activity and self-monitoring (n = 380) are addressed together.

#### 3.4.3. Mode of Delivery

*Support delivery methods.* In the intervention arms, half of the arms (53%) used support sessions; 10% used clinical visits, 11% were self-guided, and a quarter (26%) of the interventions used a combination of methods. Almost half of the interventions were conducted face-to-face; one-third of the interventions used a combination of face-to-face contacts and remote mediums, mainly phones (Table 2). Two-thirds of the interventions were given to individual patients. One-third in groups (not in table).

*Type of location.* Most interventions took place in a single location (75%). Outpatient care (43%) and homecare (24%) were the locations mentioned most often; 16% of the interventions took place in a virtual surrounding; 15% in community settings; SMIs for diabetes were hardly given in hospitals, long-term care facilities or at the workplace (Table 2).

*Type of provider.* In the majority of the interventions (58%), only one provider was involved in the intervention arm, most of the time being a nurse (36%), educator (29%), physician (20%) or nutritionist (18%). Peers and laypersons, psychologists or social workers were hardly involved in SMIs for T2DMs. In one-third of the intervention arms, two or more providers were involved. 

*Total time devoted and intensity.* Information on the duration or intensity of the interventions was missing in the majority of the included studies, and there was a huge variation in the way these variables were reported across studies. Due to the large amount of missing data regarding the exact intensity of the different contacts with patients, we made an assessment of each study arm on whether the amount of contact between the patient and provider(s) added up to at least 10 h, which we labeled as *high* intensity, or not, which we labeled as *low* intensity. Based on all data available, 67% of the intervention arms were considered to be low intensity. 

### 3.5. Outcomes Reported in the Included RCTs 

Table 3 shows the frequency of reported outcomes in the 665 included RCTs. Clinical outcomes were most frequently used as outcomes for the effectiveness of SMIs in T2DM, including HbA1c (83%), weight (53%), lipid profile (45%) and blood pressure management (42%). Quality of life and physical activity were reported as outcomes in 27% of the studies. Other outcomes, such as adherence to a diet or medication, were reported in less than 16% of the trials. One out of six addressed outcomes related to empowerment, such as self-efficacy (18%) or knowledge (16%). Other empowerment outcomes, such as patient activation or level of health literacy, were not used as outcome measures. The same counts for outcomes related to experiences with care and healthcare use (<5%).

Figure 5 presents the combinations of self-management support techniques and outcomes from the COS. To improve HbA1c, education was most often used, followed or combined with self-monitoring and goal setting. Also, in targeting other outcomes such as self-efficacy, blood pressure and quality of life, education, monitoring and goal setting for T2DM patients are favorite. Other techniques, such as enhancing problem-solving skills, learning how to handle emotions, and shared decision-making, were used relatively less often.

### 3.6. Risk of Bias of the Included Studies

Figure 6 shows that most studies had a low risk of bias in the sequence generation of the random number for the allocation of participants, but there was a lack of clarity in reporting the methods for concealment of the allocation. The main methodological limitation of the included studies was the lack of blinding of the intervention. This limitation affected the assessment of the subjective outcomes (i.e., quality of life) and objective outcomes that might be influenced by the assessor (i.e., blood pressure). Around 40% of the studies also have a significant number of drop-outs during follow-up, raising concerns about the high risk of bias due to attrition in those studies. The risk of selective reporting was more difficult to evaluate as few studies made available their protocols before the publication of the results. 

## 4. Discussion

Using evidence mapping, this study provides a systematic overview of the design and content of RCTs on SMIs for adults living with T2DM and targeting patient-relevant outcomes. SMIs published between 2010 and 2018 are described in terms of their content and main characteristics, including self-management support techniques, targeted behaviors, mode of delivery, and the extent to which patient-relevant outcomes were measured.

### 4.1. Main Findings

We identified 665 RCTs, of which a third came from the United States and 21% from a European country. Although the attention paid to research on self-management in Europe has increased steadily during recent years, Europe still seems to lag behind other parts of the world, such as the US, Asia, and Australia.

From the descriptives of the intervention arms, it became clear that SMIs for T2DM are multi-component complex interventions: the number of self-management support techniques varied widely, with sharing information, self-monitoring, goal setting, and enhancing problem-solving as the most often used techniques, alone or in combination. Likewise, most studies targeted a number of expected behaviors, often simultaneously, including self-monitoring, healthy eating, being physically active, and condition-specific behaviors like checking one’s feet and medication use. Most of the intervention arms used support sessions with face-to-face contacts, often combined with contacts by phone; they took place in a single location and were given by one or two providers; only 16% of the SMI’s interventions were organized online, in a virtual surrounding despite the fact that digital interventions have a huge potential to improve outcomes, at least for HbA1C [20,21]. SMIs for diabetes took mainly place in outpatient care or at home and were hardly given in hospitals, long-term care facilities or at the workplace. We found a very limited number of RCTs taking into account social vulnerability aspects (i.e., socio-economic status, minority groups or health literacy), and those that did mainly came from the US. Also, in Europe, social inequalities in people living with T2DM are important as European populations are ethnically and culturally diverse due to international migration. Evidence indicates large ethnic inequalities in the prevalence of T2DM, and there is a clear need for investment in research among migrant populations in Europe to gain insight into factors driving the high burden of type 2 diabetes in these groups and to facilitate prevention and treatment efforts, tailored to their specific needs [22].

An important finding from this study is that the outcomes relevant from a patient’s perspective (COS) were unequally addressed in the studies found: clinical outcomes formed the large majority with few considerations of quality of life and barely any consideration of outcomes to improve empowerment (e.g., self-efficacy, knowledge, health literacy), satisfaction with care and healthcare use. This finding is in line with a recent Danish study that came to the same conclusion [23]. With the aging of the population and an increasing number of older people with one or more chronic conditions, person-centered interventions that address the needs and preferences of older people and add to active and healthy aging are important and highly advocated at the EU and global level [24]. This study shows that the studies found hardly addressed the outcomes that empower patients, such as health literacy, self-efficacy, knowledge and interaction with care professionals. In line with this, it was striking to see that among the self-management techniques reported to support self-management, shared decision-making was only mentioned explicitly in 5% of the studies. Interviews during the COMPAR-EU project with older T2DM patients highlighted that patients often feel stigmatized and have the feeling that they do not receive an equal change in the shared decision-making process because they are too old or too frail [25]. How to optimally involve patients in shared decision-making on an equitable basis needs further attention.

### 4.2. Study Findings in the Context of Other Research

Currently, as far as we know, there is no systematic evidence on the content and design of self-management interventions for T2DM comparable to the way we produced them in this study. Several systematic reviews have summarized or studied specific components of SMIs separately, for example, self-management educational programs [26], lifestyle modification, self-management education and patient empowerment [27], but not in an integrated way as we did by using evidence mapping and following a taxonomy. Regarding the reporting of interventions, we found much missing data on the exact way interventions were designed, the intensity of interventions, and how outcomes in studies were operationalized and measured. These findings are in agreement with the findings of comparable studies in COPD and obesity that were conducted within COMPAR-EU and also found gaps in the reporting of mode of delivery, intensity, location and providers involved and a huge variety in the way these variables were reported across studies [28,29]. This study showed that 20% of the SMIs were tailored, which means that the content of the intervention or the way it was delivered was adapted to the study population. This percentage is higher than the percentage of tailored studies on SMIs in COPD and obesity, which were below 5% [28,29] and points to the fact that diabetes is generally a precursor in self-management research as compared to other chronic conditions [20]. Tailoring interventions in both design and content is a way to realize more person-centered care [24].

### 4.3. Limitations and Strengths of This Study

A number of strengths and limitations should be mentioned. The strengths of this study are a very large set of RCTs, a rigorous methodology and the use of a taxonomy and set of outcomes that was developed together with patients, professionals and implementation experts, guaranteeing the relevance of our findings for practice. A limitation of this study was that the selected interventions included the period until 2018. It is possible that some of the areas with limited information reported have improved in recent years. However, in some cases, there were so few studies (e.g., health literacy, minority groups) that it is likely that the detected gaps will still be valid. This is confirmed in a number of recent systematic reviews of self-management interventions in T2DM that included studies of a more recent date. They also stress the lack of patient-relevant outcomes in studies evaluating SMIs in T2DM [23], the heterogeneity in design, and the lack of sufficient detail in intervention descriptions, which complicates the evaluation of these interventions and the use of these interventions in clinical practice [30,31]. However, an update of this study in a few years would be valuable, given the speed at which SMIs are developing.

### 4.4. Implication for Practice and Research

The current study provided a comprehensive overview of RCTs of SMI for patients with T2DM but also highlighted the heterogeneity and gaps that exist in both the design and reporting of these interventions. Including intervention descriptions and design characteristics in the areas with less information detected could help to obtain a better picture of the reported results of SMIs, their value and how they can be replicated. This is especially important as shortcomings in reporting and design even have a more detrimental effect on the building of solid evidence when interventions are complex, such as is the case in SMIs. 

During the past decade, increased focus has been placed on the importance of improving chronic care delivery through the measurement of patient-important health outcomes. Care should be person-centered, respectful of and responsive to a person’s preferences, needs, and values, and ensuring that these guide all clinical decisions. Shared decision-making is the method par excellence to realize person-defined care, even in an older population such as T2DM. Our findings showed that there is much win in this respect. For implementation efforts, it is also important to take patient-relevant outcomes into account and tailor intervention content to the needs of specific groups (e.g., minority groups, people with low SES or health literacy) and not use a one-size-fits-all approach.

In addition, future research should also explore the content and design of SMIs focusing on specific subgroups of T2DM patients that have been marginally addressed or excluded in RCTs so far: for example, T2DM patients with comorbidities. Diabetes patients who suffer from comorbidity or diabetes as comorbidity by other chronic diseases complicate self-management for patients to a great extent, and specific designs may be needed to support people with comorbidities. Now, people with comorbidities are often excluded in randomized controlled trials to evaluate self-management interventions, but this does not correspond with the reality that over 75% of diabetes patients suffer from co-morbidity [32]. In addition, the COMPAR-EU T2DM panel also suggested that SMIs tailored to frequently occurring population characteristics in T2DM other than comorbidity (e.g., culture, limited HL) may also increase equity and acceptability when implementing SMIs [31]. 

Finally, regarding the design of the studies, all included studies in this study followed a golden standard study design—RCT, to evaluate the effectiveness of SMIs. However, results show that most studies had a high-performance bias, and some had detection and attrition bias. Considering the high costs of the RCT and the difficulty of blinding participants for SMI research, future studies may consider focusing on more realistic reviews to explore the effectiveness of SMIs. Although the current study did not focus on the description of the usual care arms that were used in many studies as comparators, we encountered huge gaps in the reporting of usual care, which blurred the effectiveness of SMIs and hindered the final decision-making. Future studies should also clearly explain the components of usual care arms in the SMI studies to better reflect the effectiveness of SMIs.

## 5. Conclusions

Although the current evidence for SMIs in T2DM patients is enormous, gaps and disparities exist in this field, including the inconsistency of SMI components used and poor reporting, lack of attention to the diversity among T2DM patients and a lack of addressing patient-relevant outcomes. More standardized and streamlined research is needed to better support research on the effectiveness and cost-effectiveness of SMIs, facilitate implementation efforts and benefit patient care.

## Figures and Tables

**Figure 1 healthcare-11-03156-f001:**
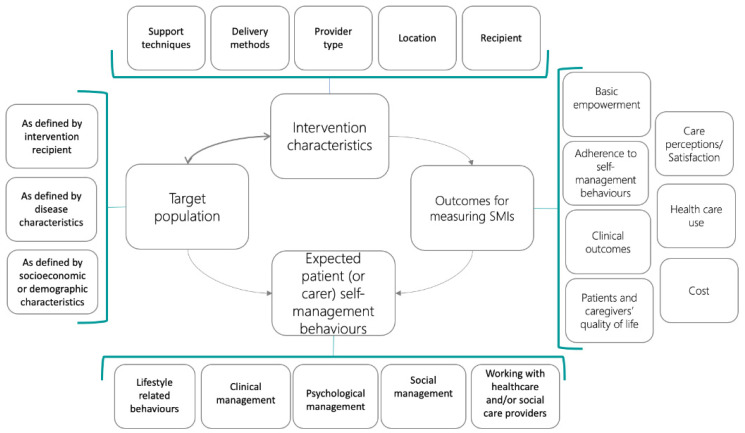
COMPAR-EU self-management intervention taxonomy (Source: Orrego, Health Expect. 2021).

**Figure 2 healthcare-11-03156-f002:**
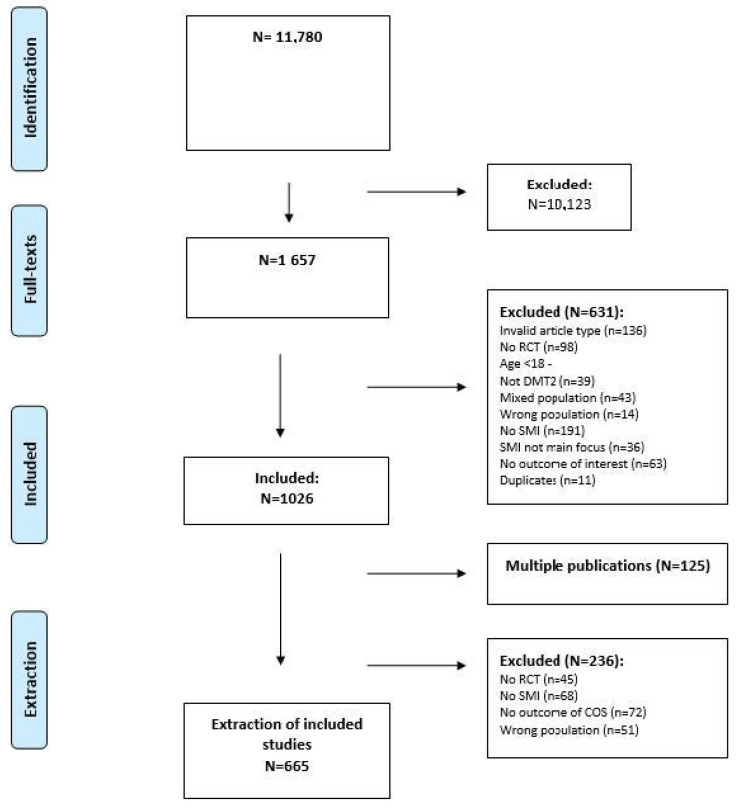
Preferred Reporting Items for Systematic Reviews and Meta-Analyses (PRISMA) flow chart.

**Figure 3 healthcare-11-03156-f003:**
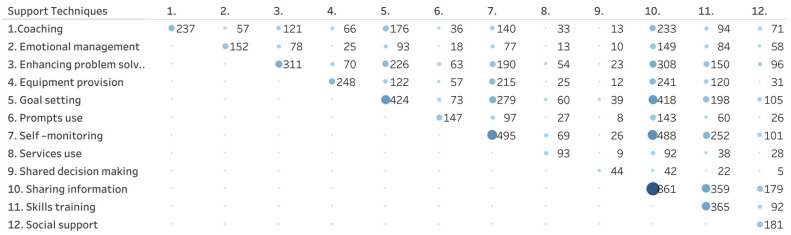
Frequency in which support techniques are combined across intervention arms (n = 879). The size and color of the bubble indicates the number of studies including each combination presented, with bigger size and darker color referring to more studies included.

**Figure 4 healthcare-11-03156-f004:**
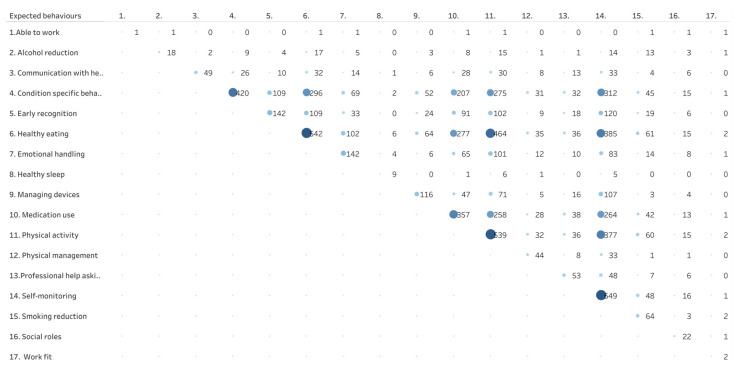
Frequency in which expected behaviors are combined across intervention arms (n = 879). The size and color of the bubble indicates the number of studies including each combination presented, with bigger size and darker color referring to more studies included.

**Figure 5 healthcare-11-03156-f005:**
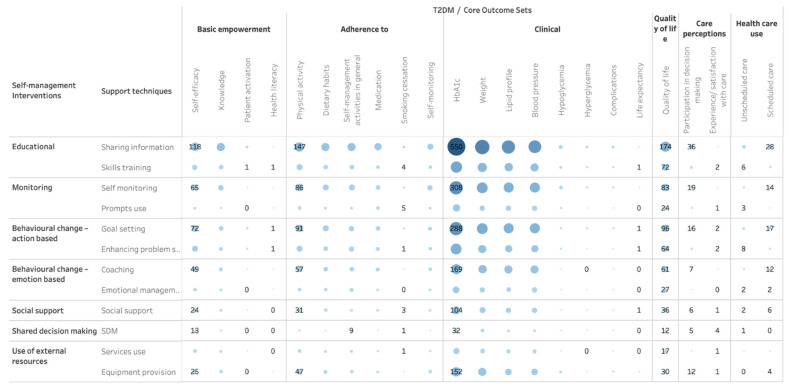
Support techniques for SMIs used to address outcomes for T2DM. The size and color of the bubble indicates the number of studies including each combination presented, with bigger size and darker color referring to more studies included.

**Figure 6 healthcare-11-03156-f006:**
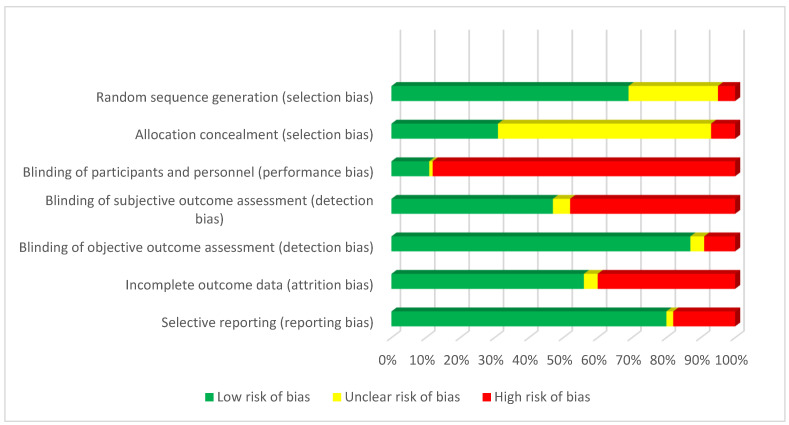
Risk of bias of included RCTs.

**Table 1 healthcare-11-03156-t001:** Characteristics of included RCTs (N = 665) and population characteristics (N = 164,437).

Characteristics of Included RCTs		N (%)
**Country (Top 5)**	US	233 (35%)
	Iraq	47 (7%)
	United Kingdom	41 (6%)
	China	37 (6%)
	Korea	30 (5%)
**Type of population**	Patient	655 (99%)
	Caregiver	1 (0%)
	Mixed	8 (1%)
**Type of comparison**	Head-to-head interventions	135 (20%)
	Intervention(s) vs usual care	530 (80%)
**Type of usual care (n = 530)**	Usual care	367 (69%)
	Usual care plus	163 (31%)
**Number of arms across all interventions (n = 665)**	2	598 (90%)
	3	56 (8%)
	4	10 (2%)
	5	1 (0%)
**Unit of allocation**	By individual	612 (92%)
	At population level	53 (8%)
**Specific target population**	T2DM with low SES	72 (11%)
	T2DM from minority groups	89 (13%)
	T2DM with low Health Literacy	29 (4%)
	T2DM with comorbidities	88 (13%)
**Characteristics of the total population included across interventions**		**Mean (SD)**
% women Mean age		57.0 (49.0–67.0) 57.6 (54.3–61.5)
Mean number of years since diagnosis		8.6 (6.6–10.8)
Mean HbA1c		8.2 (7.6–8.9)

**Table 2 healthcare-11-03156-t002:** Descriptives of intervention arms (N = 879).

Support Techniques	Type	n	%
**Educational**	Sharing information	861	98%
	Skills training	365	42%
**Monitoring**	Self-monitoring	495	56%
	Prompt use	147	17%
**Behavioral change-action based**	Goal setting	424	48%
	Enhancing problem-solving	311	35%
**Behavioral change-emotion based**	Coaching	237	27%
	Emotional management	152	17%
**Social support**	Using social support	181	21%
**Shared Decision Making (SDM)**	SDM	44	5%
**Use of external resources**	Service use	93	11%
	Equipment provision	248	28%
**Expected self-management behaviors**	**Type**	**n**	**%**
**Lifestyle-related**	Healthy eating	542	62%
	Physical activity	539	61%
	Smoking reduction	64	7%
	Alcohol reduction	18	2%
	Healthy sleep	9	1%
**Clinical management**	Self-monitoring (e.g., blood sugar)	549	63%
	Condition-specific behavior	420	48%
	Medication use	357	41%
	Early recognition of symptoms	142	16%
	Managing devices	116	13%
	Professional help-seeking	53	6%
	Physical management	55	5%
**Psychological management**	Handling emotions	142	16%
**Social management**	Fulfilling social roles	22	3%
	Able to work/being fit	3	1%
**Experience with care**	Communication with healthcare	49	6%
**Type of encounter**		**n**	**%**
	Clinical visits	85	10%
	Support sessions	465	53%
	Self-guided	98	11%
	Combination	229	26%
**Way of delivery**		**n**	**%**
	Face-to-face	395	45%
	Remote	181	21%
	Combination	293	33%
**Location**		**n**	**%**
	Outpatient care	376	43%
	Homecare	207	24%
	Community care	134	15%
	Primary care	143	16%
	Virtual/online	142	16%
	Other (hospital, work, long-term care)	1–11	<2% each
**Type of provider**		**n**	**%**
	Nurse	309	35%
	Physician	177	20%
	Educator	258	29%
	Nutritionist	162	18%
	Other		<9%
**Intensity**		**n**	**%**
	Low intensity (<10 h)	592	67%
	High intensity (≥10 h)	286	33%

**Table 3 healthcare-11-03156-t003:** Frequency in which Core Outcomes (COS) were addressed in studies (N = 665).

Category of Outcomes	Type	n	%
**Basic empowerment**	Self-efficacy	120	18%
	Knowledge	109	16%
	Patient activation	11	2%
	Health literacy	3	0%
**Adherence to**	Physical activity	153	23%
	Dietary habits	106	16%
	Self-management activities in general	111	17%
	Medication	92	14%
	Smoking cessation	14	2%
	Self-monitoring	63	10%
**Clinical**	HbA1c	550	83%
	Weight	353	53%
	Lipid profile	296	45%
	Blood pressure	281	42%
	Hypoglycemia	28	4%
	Hyperglycemia	13	2%
	Complications	15	2%
	Life expectancy	2	0%
**Quality of life**	Quality of life	180	27%
**Care perceptions**	Participation in decision-making	4	1%
	Experience/satisfaction with care	34	5%
**Health care use**	Unscheduled care	26	4%
	Scheduled care	13	2%

## Data Availability

Data (transcripts) from the COMPAR-EU project have been deposited according to the data management plan at NIVEL. The qualitative datasets used and/or analyzed during the current study are available from the corresponding author upon reasonable request, provided they do not identify interviewees.

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
