# Peer review of "Self-Management Interventions for Adults Living with Type II Diabetes to Improve Patient-Important Outcomes: An Evidence Map"

_healthcare, 2023, doi:10.3390/healthcare11243156_

Round 1

Reviewer 1 Report

Comments and Suggestions for Authors

The review summarizes the intervention components and characteristics in randomized controlled trials (RCTs) related to T2DM. The success of T2DM management largely depends on the patient's ability to accept and take responsibility for his disease. In addition to medication, proper nutrition, and lifestyle, self-management plays an important role. Most of the outcomes of self-management interventions were of clinical significance, despite the importance of non-clinical - such as management of emotions, shared decision-making, and minority affiliation - linked to these outcomes by patients. The authors aimed to "summarize intervention components in RCTs related to self-management for T2DM and identify components that are insufficiently incorporated or insufficiently reported into the design of the intervention."

The authors highlighted person-centered care (alias personalized medicine), from which all decision-making processes in clinical practice should be guided. They also highlighted the importance of shared decision-making, but on the other hand, point to the prevalence of T2DM in an aging, albeit increasingly computer-literate population, but also in the population of minority groups (e.g. immigrants), in whose case their social status does not allow online sharing of their health status. In the end, we learned the shortcomings and the gaps in both the design and reporting of the interventions, but unfortunately, these factors will continue to be limited in the most vulnerable population groups.

I am not quite sure whether the authors have fulfilled the goal stated in the introduction of the work. From this point of view, I would expect at least a hint of proposed solutions to improve the self-monitoring interventions that will help the most vulnerable patients.

The review contains an inaccuracy that should be corrected:

line 63: imprecise definition of T2DM - not just due to insulin shortage, but often due to resistance at a normal insulin production (peripheral insulin resistance, impaired regulation of hepatic Glc production, declining beta-cell function/failure...) 

Author Response

Please find below our answers to your comments:

I am not quite sure whether the authors have fulfilled the goal stated in the introduction of the work. In the end, we learned the shortcomings and the gaps in both the design and reporting of the interventions, but unfortunately, these factors will continue to be limited in the most vulnerable population groups. From this point of view, I would expect at least a hint of proposed solutions to improve the self-monitoring interventions that will help the most vulnerable patients.

Answer: The goal of this paper was to summarize intervention components in RCTs related to self-management for T2DM and identify components that are insufficiently incorporated or insufficiently reported into the design of the intervention and as such hinder the interpretation of findings or implementation of an SMI in the own context. By detecting shortcomings either in the design or in the description of SMIs for T2DM, we aimed to provide a foundation for the design of new/better interventions in diabetes self-management, make replication/ implementation easier and contribute to the improvement of the future reporting of interventions [line 86-90]. So, our primary focus was not on vulnerable groups. However, one of our findings was that information about the population characteristics was hardly provided  - e.g. hardly no information on SES, ethnic background or health literacy level, and that only a minority of studies was tailored. This information is of utmost important when one  wants to interpret findings,  implement SMIs in the own context and for being able to estimate whether the content of the SMI in the RCT fits the needs of the local population in the own country. Given the high prevalence of T2DM in minority groups (see also line 364-365)  and  low SES groups and the importance of HL and tailoring for successful self-management (support) we highlighted these omissions in the discussion and add a sentence that SMI should not use a one size fits all approach but take into consideration the specific needs of more vulnerable groups [line 435-436].

The review contains an inaccuracy that should be corrected: line 63: imprecise definition of T2DM - not just due to insulin shortage, but often due to resistance at a normal insulin production (peripheral insulin resistance, impaired regulation of hepatic Glc production, declining beta-cell function/failure...)

Answer: Thanks for pointing out to this. We adapted the definition of T2DM in line with your comments in line 63.

Reviewer 2 Report

Comments and Suggestions for Authors

Several self-management intervention approaches exist to improve and control glycaemic levels in patients with type 2 diabetes (T2D). However, there is a lack of knowledge which approach is relatively better and more effective to treat T2DM. Yang Song and his/her colleagues tried to summarize intervention components and characteristics in randomized control trials (RCTs) and identify components that are insufficiently incorporated into the design of the intervention or insufficiently reported.

I have a few comments and questions:

I am not clear with the statement ......... (median 123, range 10-14.559). Line 37.

o   In BOX 1, what does NMAs mean?

o   Is the study a systematic review or a scoping review? It is not clear. Because you choose to use PRISMA Extension for Scoping Reviews (PRISMA-ScR) line 104/105. If you choose scoping review, why do you choose scoping review over systematic review?

o   Line 111 MEDLINE but below in line 180 it was PubMed. Be consistent!

o   In Appendix 1 the search strategy mentioned was only for PubMed, what about for others mentioned such as CINAHL, Embase, Cochrane, and PsycINFO?

o   On the PRISMA, how many studies extracted from each search engine (PubMed, Embase, CINAHL, PsycINFO, and Cochrane) were not mentioned. It would be nice to include it.

o   Quality assessment was not carried out for the selected articles. For example, Newcastle-Ottawa Scales is used to assess the quality of selected articles.

o   The authors mentioned only a few biases. What about other sources of bias such as publication bias that may affect your study?

o   Table 1, recheck the percentages, some percentages are not correct. For instance, the US is 35.04, not 34% and Usual care is 55.2%, not 770%.................

o   In Table 1, why do you use median (IQR) to report % sex? Age, HbAlC, and number of years since diagnosis were reported in means but you put them as median (IQR). Please correct it.

o   Could you please put the list of included studies on an Excel sheet or other files as a supplement file? Readers may be interested in looking at the included studies. 

Author Response

Dear reviewer,

Thank you very much for your usefull comments. Please find our answers below:

  • I am not clear with the statement ......... (median 123, range 10-14.559). Line 37. Answer: It is saying that across all studies together 164437 patients participated with a median value of 123 across studies and varying between 10 as the lowest and 14559 as the highest number of participants within studies. A median value means that half of the studies has 123 or less participants and the half more than 123.
  • In BOX 1, what does NMAs mean? Answer: NMA means Network Meta Analysis. We clarified this in Box 1
  • Is the study a systematic review or a scoping review? It is not clear. Because you choose to use PRISMA Extension for Scoping Reviews (PRISMA-ScR) line 104/105. If you choose scoping review, why do you choose scoping review over systematic review? Answer: The article is an evidence and gap map review. This type of review, like scoping reviews or mapping reviews are evidence synthesis methodologies that address broad research questions, aiming to describe a bigger picture than systematic reviews. Scoping reviews and evidence and gap maps have similarities that unite them as a group but also have unique differences. Scoping reviews are typically more exploratory than mapping reviews and EGMs, not requiring an a priori set of codes in order to describe data and may draw upon a range of sources of information. For the reporting of evidence and gap map review the PRISMA Extension for Scoping Reviews is suggested. We have clarified this further in the manuscript. See also the following references: Campbell, F., Tricco, A.C., Munn, Z. et al. Mapping reviews, scoping reviews, and evidence and gap maps (EGMs): the same but different— the “Big Picture” review family. Syst Rev 12, 45 (2023). https://doi.org/10.1186/s13643-023-02178-5  and Munn, Z., Peters, M.D.J., Stern, C. et al. Systematic review or scoping review? Guidance for authors when choosing between a systematic or scoping review approach. BMC Med Res Methodol 18, 143 (2018). https://doi.org/10.1186/s12874-018-0611-x
  • Line 111 MEDLINE but below in line 180 it was PubMed. Be consistent! Answer: adapted; we replaced MEDLINE by PubMed.
  • In Appendix 1 the search strategy mentioned was only for PubMed, what about for others mentioned such as CINAHL, Embase, Cochrane, and PsycINFO? Answer: we have added the searches for the other search strategies in Appendix 1. Please see attachement
  • On the PRISMA, how many studies extracted from each search engine (PubMed, Embase, CINAHL, PsycINFO, and Cochrane) were not mentioned. It would be nice to include it. Answer: the number of studies on each database can be found in the new Appendix 1 ( PubMed 6.988; Embase 2.602; PsycINFO 826; CINAHL 2.229 and Cochrane 7.386). We merged them all into 1 database (n=20.031) and then removed the duplicates resulting in 11.780 references.
  • Quality assessment was not carried out for the selected articles. For example, Newcastle-Ottawa Scales is used to assess the quality of selected articles. Answer: We are not sure what the reviewer is referring to. Risk of bias was assessed using Cochrane's tool for assessing risk of bias in included studies, and then a second reviewer verified the judgments. We did not use the Newcastle-Ottawa scale.
  • The authors mentioned only a few biases. What about other sources of bias such as publication bias that may affect your study? Answer: In this evidence map we report all the domains included in the Cochrane risk of bias tool, these include selection, reporting, detection attrition and reporting. This tool does not include the risk of bias due to missing results (publication bias), as it is an aspect that relates to the synthesis rather than study level. Furthermore, given the limitations of funnel plots, we decided not to include this aspect in the evidence map.
  • Table 1, recheck the percentages, some percentages are not correct. For instance, the US is 35.04, not 34% and Usual care is 55.2%, not 770%................. Answer: our apologies for these carelessness; we adjusted them.
  • In Table 1, why do you use median (IQR) to report % sex? Age, HbAlC, and number of years since diagnosis were reported in means but you put them as median (IQR). Please correct it. Answer: the reviewer is right. We corrected it. It are all means and SD
  • Could you please put the list of included studies on an Excel sheet or other files as a supplement file? Readers may be interested in looking at the included studies. Answer: we added a supplementary file. Please see attachement

Reviewer 3 Report

Comments and Suggestions for Authors

Self-management interventions for adults living with type II diabetes to improve patient-important outcomes: evidence map.

I appreciate the opportunity to review this paper on examining Self-management interventions for adults living with type II diabetes to improve patient-important outcomes: an evidence map.

General comments

Paper addresses an important aspect of Health, which   is Self-management interventions for adults living with type II diabetes to improve patient-important outcomes: an evidence map.

Several studies have explored different aspects of the effects of type 2 diabetes. This paper is an addition with context relevance for Self-management of type two diabetes, hence, I consider it appropriate for publication. However, Additional revision and proof reading is recommended to improve the content, formatting, and language of the paper.

Abstract

The abstract is well structured and descriptive, background of abstract clearly indicate the need for Self-management interventions for adults living with type II diabetes to improve patient-important outcomes. Findings are specific to the topic under study, and recommendations were given to adequately align with findings. However, the reader may not have any knowledge of the COMAPAR-EU project mentioned in the abstract by the authors. Hence, I suggest that the full meaning of the COMAPAR-EU project should be captured in the abstract. Also on line 44, the authors wrote that; patient activation or satisfaction with care ‘’where’’ hardly used as outcome ‘Where’ was used instead of ‘were.’’

Introduction/Background

The background was well written and the need for the study justified.

Objectives

The objectives of the study were explicitly stated.

Methods

The methodological orientation of the study was well described. Researchers vividly described the methodology for conducting the research.  

Criteria for determining sample size, and the formular for screening, extracting e and collecting data were explicitly described.

Data Analysis

A description of the process of data analysis, and statistical tests used for the analysis were described, however, measures for ensuring validity and reliability of the research were not vividly stated.

Ethical consideration

Ethical considerations were NOT vividly spelt out by the researchers.

Findings

Participants’ characteristics were described, self-management practices and expected self-management behaviours of adults living with Type 2 Diabetes Mellitus were well spelt out in the findings.

Discussion

The discussion compared findings with other studies. The trend of discussion clarified self-management practices and expected self-management behaviours of people living with type 2 Diabetes Mellitus. They also provided thorough interpretation of findings, relating it to the topic under study to further enhance the discussion.

Limitations

The limitations and strengths of the study were outlined by the researchers.

Implications for nursing practice

Implications were well-articulated and properly aligned to findings.

Conclusions are drawn from the study data or findings.

References

References have been cited. It is recommended that author(s) ensure compliance with the journal requirement. 

Comments on the Quality of English Language

Additional revision and proof reading is recommended to improve the content, formatting, and language of the paper.

Author Response

We would like the reviewer for the constructive remarks. We answered below point by point

  • Additional revision and proof reading is recommended to improve the content, formatting, and language of the paper. Answer: we asked a native English speaker to proofread for English language. The formatting of Tables etc is automatically generated by the journal.
  • Abstract- the reader may not have any knowledge of the COMPAR-EU project mentioned in the abstract by the authors. Hence, I suggest that the full meaning of the COMAPAR-EU project should be captured in the abstract. Answer: We agree that introducing COMPAR-EU already in the abstract without further explanation is a little confusing. We decided to remove COMPAR-EU from the abstract and only introduce COMPAR-EU in the method section as we feel it is not necessary to already mention this project in the abstract.
  • Also on line 44, the authors wrote that; patient activation or satisfaction with care ‘’where’’ hardly used as outcome ‘Where’ was used instead of ‘were.’’ Answer: adapted
  • A description of the process of data analysis, and statistical tests used for the analysis were described, however, measures for ensuring validity and reliability of the research were not vividly stated. Answer: the review methodology was established before data extraction and extensively described and registered in PROSPERO (CRD42020155441) and published (see line 107-108). Therefore we decided not to go in too much detail again. However as we used a data extraction form based on a highly valued taxonomy agreed upon by professionals and patients with clear descriptions of the different SMI components, calibration processes before starting the abstract and full text screening for all reviewers , and as each extraction was checked by another independent reviewer we believe our research was of high quality.
  • Ethical considerations were NOT vividly spelt out by the researchers. Answer: as far as we are aware of, reviews do not need any ethical approvals or informed consent.
  • References have been cited. It is recommended that author(s) ensure compliance with the journal requirement. Answer: we checked and adapted where necessary